# One-Step Lignin Refining Process: The Influence of the Solvent Nature on the Properties and Quality of Fractions

**DOI:** 10.3390/polym14122363

**Published:** 2022-06-11

**Authors:** Oihana Gordobil, René Herrera Diaz, Jakub Sandak, Anna Sandak

**Affiliations:** 1InnoRenew CoE, Livade 6, 6310 Izola, Slovenia; renealexander.herrera@ehu.eus (R.H.D.); jakub.sandak@innorenew.eu (J.S.); anna.sandak@innorenew.eu (A.S.); 2Department of Chemical and Environmental Engineering, University of the Basque Country (UPV/EHU), Plaza Europa, 1, 20018 Donostia-San Sebastian, Spain; 3Faculty of Mathematics, Natural Sciences and Information Technologies, University of Primorska, Glagoljaška 8, 6000 Koper, Slovenia; 4Andrej Marušič Institute, University of Primorska, Titov trg 4, 6000 Koper, Slovenia

**Keywords:** hardwood kraft lignin, one-step fractionation, organic solvents, hygroscopicity, thermal properties, antioxidant capacity

## Abstract

Heterogeneity of kraft lignin is one of the main limitations for the development of high-performance applications. Therefore, refining lignin using organic solvents is a promising strategy to obtain homogenous fractions with controlled quality in terms of structure and properties. In this work, one-step refining processes for hardwood kraft lignin using nine organic solvents of different chemical nature and polarity were carried out with the aim of investigating and understanding the effect of the type of organic solvent on the quality of resulting fractions. Structural features of both soluble and insoluble lignin fractions were assessed by GPC, Py-GC-MS, and FTIR linked to PCA analysis. Moreover, functional properties such as physical appearance, hygroscopicity, antioxidant capacity, and thermal properties were evaluated. The results evidenced the relationship between the nature and polarity of the solvents and the properties of the obtained soluble and insoluble fractions.

## 1. Introduction

The global interest in lignin valorization is driven by its industrial availability since it is generated in large quantities from cellulose pulp production (80% of the available lignin) as an affluent, or as a by-product from other pre-treatment processes of lignocellulosic biomass [1]. Currently, pulp and paper industries produce 50–70 million tons of lignin annually, and an increase in production of up to 225 million tons per year by 2030 [2] is estimated. In recent years, research on the diverse functional and bioactive properties of lignin polymer has led to increased interest in this bio-sourced compound, for several applications. Lignin has the potential to become a leading renewable material feedstock to overcome dependence on fossil-based resources, and its valorization will play a vital role in the biorefinery industry for the production of a bio-based platform of chemicals, fuels, and materials with huge applicability in the polymer, cosmetic and personal care, biomedical and pharmaceutical, and energy sectors [3,4,5]. The control of key parameters during lignin extraction or recovery determines the quality of the resultant lignin polymer in terms of purity, molecular characteristics, and properties. However, lignin is an extremely complex natural polymer with high chemical variability, and after extraction, an additional conversion process is usually required to overcome the technical issues related to its heterogeneity and presence of impurities [6]. Lignocellulosic-based biorefineries require consistent quality in terms of chemical structure and product characteristics (appearance, purity), as well as in terms of sustainability and economic efficiency, in order to compete with petroleum-based products. Therefore, studies of methodologies to process and obtain products from lignin are necessary to achieve high performance in specific applications for the market. To date, lignin fractionation has been investigated by different approaches, such as selective precipitation [7], membrane-mediated processes [8,9], and extraction with organic solvents [10,11]. The use of organic solvents is one of the most reported methods, and several authors have studied the fractionation of lignin by solvent mixtures and sequential cascade using a wide range of organic solvents such as alcohols, ketones, esters, ethers, chlorinated solvents, and alkenes, achieving lignin fractions with higher homogeneity [12,13,14]. However, the relationship between the nature of the solvent and the chemical structure of the extracted lignin is still unknown. Additionally, recently published techno-economic analyses related to lignin fractionation revealed high capital expenditures (USD 20 to 32 million) and operational costs based on the solvent choice, ranging from 390 to 7280 USD/t [15,16]. Therefore, in this context, one-step fractionation gains strength from the industrial point of view as the most feasible solution to minimize both operating and capital cost. In this work, one-step fractionation of industrially produced hardwood kraft lignin (HKL) was investigated. Nine organic solvents of different chemical nature and polarity were selected, and the resulting soluble and insoluble fractions from each refining process were assessed in terms of molecular weight, structural properties, and other properties such as hygroscopicity, antioxidant capacity, and thermal properties. This study discusses the potential valorization routes of lignin fractions according to their structural and functional properties.

## 2. Materials and Methods

### 2.1. Materials

Hardwood kraft lignin (HKL) was isolated by precipitation from industrial black liquor produced during the manufacture of cellulose pulp from Eucalyptus chips. For the precipitation, sulfuric acid (98%) was used as an acidifying agent to lower the pH to 2. The precipitated HKL was recollected by filtration, washed until neutral pH was achieved, and dried at controlled room conditions (25 °C). The resulting lignin was manually ground with a mortar and stored for further analysis. The yield of precipitated lignin from the industrial black liquor was 32 g/L. The organic solvents used for the fractionation process were methanol (99.8%, Honeywell, Charlotte, NC, USA), ethanol (absolute, Honeywell, Charlotte, NC, USA), 2-propanol (99.8%, Merck, Ljubljana, Slovenia), acetone (Merck, Ljubljana, Slovenia), ethyl acetate (Merck, Ljubljana, Slovenia), dichloromethane (DCM) (ACS reagent, Carlo Erba, Emmendingen, Germany), diethyl ether (Merck, Ljubljana, Slovenia), hexane (Carlo Erba, Emmendingen, Germany), and petroleum ether (Merck, Ljubljana, Slovenia). Other solvents and reagents used for lignin characterization were purchased from Sigma Aldrich (Merck, Ljubljana, Slovenia) *N*,*N*-Dimethylformamide, lithium bromide, dimethyl sulphoxide (99.9%), Folin–Ciocalteau reagent, sodium carbonate (99.8%) DPPH, and ascorbic acid.

### 2.2. One-Step Fractionation of Hardwood Kraft Lignin

Lignin fractions were extracted from the crude hardwood kraft lignin (HKL) using organic solvents in a one-step fractionation process to evaluate the influence of the type of organic solvent on lignin chemistry. For this purpose, three polar protic solvents (methanol, ethanol and 2-propanol), three polar aprotic solvents (acetone, ethyl acetate and dichloromethane), and three non-polar solvents (diethyl ether, hexane and petroleum ether) were selected. Kraft lignin (1 g) was suspended in 50 mL of solvent with constant stirring (1000 rpm) at room temperature for 2 h. The soluble and insoluble fractions were separated by filtration. Insoluble fractions were washed and dried overnight in vacuum at 30 °C, while soluble fractions were recovered by removing the solvent under reduced pressure in a rotary evaporator. All the organic solvents used have a low boiling point temperature (35–82 °C), which allowed for the easy recovery of the solvent after each fractionation process. Each process was carried out in triplicate to check the replicability of the single-fractionation process. The solubility yields were calculated on a dry basis. Finally, soluble and insoluble fractions were stored at 25 °C and 0% RH to avoid moisture interferences in further analysis. The Pearson correlation coefficient was calculated from the data set of yields, average molecular weight, and polarity of solvents using OriginPro 2015 software.

### 2.3. Physical Appearance of Lignin Fractions

The micromorphology of crude HKL and lignin fractions were scrutinized under a digital microscope (Keyence VHX-6000, Ozaka, Japan) at 500X magnification with Epi-illumination light mode.

### 2.4. Chemical Structure Characterization

The molecular weight-average (Mw), number-average (Mn), and polydispersity (Mw/Mn) of kraft lignin (HKL) and refined lignin fractions were determined by gel permeation chromatography (GPC) (Jasco LCNet II/ADC), equipped with an RI-2031 Plus Intelligent refractive index detector, PolarGel-M column (300 mm, 7.5 mm), and PolarGel-M guard (50 mm, 7.5 mm). For the test, 0.25 mg of the sample was dissolved in 5 mL of *N*,*N* dimethylformamide (DMF) with 0.1% lithium bromide, and 20 μL of the solution was injected. The column operated at 40 °C and was eluted with *N*,*N* dimethylformamide (DMF) with 0.1% lithium bromide at a flow rate of 0.7 mL/min. Monodispersed polystyrene by Fluka was used as a standard for the calibration curve, ranging from 250 to 70.000 g/mol.

Analytical pyrolysis (Py-GC-MS) was carried out using a commercial pyrolyzer (Pyroprobe model 5150, CDS Analytical Inc., Oxford, PA, USA) and GC-MS instrument (Agilent Techs. Inc. 6890 GC/5973MSD, Santa Clara, CA, USA). Lignin samples (400–800 µg) were pyrolyzed in a quartz boat at 650 °C for 15 s at a heating rate of 20 °C/ms (ramp-off) with the interface kept at 260 °C. The pyrolysates were purged from the pyrolysis interface into the GC injector under inert conditions using helium gas. An Equity-1701(30 m × 0.20 mm × 0.25 μm) fused-silica capillary column was used. The GC oven program started with 2 min at 50°. Then it was increased to 120 °C at 10 °C/min and held for 5 min. After that, it was increased to 280 °C at 10 °C/min and held for 8 min; it was finally increased to 300 °C at 10 °C/min and held for 10 min. The identification of the pyrolysis products was accomplished by comparing their mass spectra with the National Institute of Standards Library (NIST) and with compounds reported in the literature [17,18]. Only compounds with a peak area ratio higher than 0.2% were selected for the calculation. The sum of the relevant peak areas was normalized to 100% to determine the relative abundance of compounds since the peak area is widely related to the concentration of each compound. The relative percentage of released pyrolytic products was calculated as the ratio of the sum of the areas of the peaks from the corresponding product divided by the sum of the area of all used peaks multiplied by 100%. The given results are the average of the three replicas. The infrared spectra of each lignin sample (conditioned at 25 °C and 0% RH) was scanned in triplicate in the 4000–400 cm^−1^ range at 4 cm^−1^ of spectral resolution, and accumulated an average of 64 scans with an Alpha II compact FT-IR spectrophotometer (Bruker Optik GmbH, Ettlingen, Germany) equipped with a diamond crystal with attenuated total reflectance (ATR). The obtained IR database was filtered, processed, and decomposed into a few principal components (PCA) that contained most of the variability able to determine the distribution of the samples as well as affinity of the obtained fractions. PLS-Toolbox analytical software tool (Eigenvector research incorporated) was used for signal processing and data analysis.

### 2.5. Hygroscopic Properties

Sorption and desorption properties of industrial kraft lignin and isolated fractions (soluble and insoluble) were evaluated. First, lignin samples were freeze-dried to remove any remaining moisture content due to storage. Freeze-drying is considered as a suitable technique to avoid any chemical change in the lignin chemistry and microstructure during the drying step because of the temperature [19]. Then, 0.2–0.3 g of lignin samples were conditioned in 3 L sealed containers at different relative humidity (0, 35, 75, and 95%) and at constant temperature (27 °C). An amount of 1.5 L of saturated salt/water solution was used to control the relative humidity in each container (MgCl, NaCl and water). The equilibrium mass change and moisture content were gravimetrically measured at each relative humidity to obtain experimental sorption and desorption isotherms. The sorption–desorption cycle was carried out for each lignin sample obtained from the one-fractionation process. Therefore, three replicas of each soluble and insoluble lignin type were considered for the sorption–desorption study. The data were used to calculate interpolated isotherms.

### 2.6. Total Phenolic Content (TPC) and DPPH Assay

The TPC of isolated lignins was determined by the Folin–Ciocalteau spectrophotometric method using gallic acid as a reference and dimethyl sulfoxide as a solvent, in the Mettler Toledo UV7 spectrophotometer. For the analysis, 30 µL of lignin solution (2 mg/mL), 150 µL of Folin–Ciolcalteau reagent, and 300 µL mL of Na_2_CO_3_ (200 mg/mL) were combined with distilled water to bring the volume up to 3 mL. The samples were mixed with a stirred type of vortex and kept in a thermostatic bath at 40 °C for 30 min before the spectrophotometric measurement of the absorbance at 750 nm. The total phenolic content of lignin samples was expressed as µg gallic acid (GAE) per mg of dry lignin. All parameters were calculated on a dry basis and in triplicate. The DPPH scavenging activity was assessed according to the method described by Brand-Williams et al. 1995 [20], with some modifications. DMSO was used to dissolve the lignin samples at different concentrations (0–2 mg/mL). Then, 75 µL of lignin solution was added to 2925 µL DPPH (25 mg/L in methanol). The absorbance was measured at 517 nm after 30 min of incubation at room temperature (Mettler Toledo UV7). Ascorbic acid was used as a positive control. The analyses were performed in triplicate. Each test was carried out in triplicate. The inhibition percentage of the DPPH radical was calculated according to Equation (1):(1)$$Inhibition\;\left(\%\right)=\frac{Abs_{S}-Abs_{B}}{Abs_{B}}\times 100$$
where the *Abs_S_* is the absorbance of DPPH at 517 nm in the presence of the lignin sample and *Abs_B_* is the absorbance without the addition of the antioxidant compound. The radical scavenging activity of the lignin was expressed using the term “efficient concentration” or IC_50_, which is the concentration required for 50% inhibition of the free radical.

### 2.7. Thermal Characterization

Thermogravimetric analyses were performed using thermogravimetric analyzer Discovery TGA-5500 (Waters TA Instruments, New Castle, DE, USA). A high-resolution dynamic method with a heating rate of 20 °C/min, a resolution of 4, and a sensitivity value of 1, was used for all samples. For the thermal analysis, 5–10 mg of lignin fractions were tested under a nitrogen atmosphere from 25 to 800 °C. Then, the samples were maintained under oxygen atmosphere at 800 °C for 15 min to determine the ash content. Three replicas of each lignin sample were conducted. The electro-balance was purged with nitrogen at a flow rate of 10 mL/min and the furnace at a flow rate of 25 mL/min. Thermogravimetric (TG) and derivative thermogravimetric (DTG) data generated by instruments were decoded using TA Instruments TRIOS software. DSC analyses were carried out using dynamic calorimetry system Discovery DSC-25 (Waters TA Instruments, New Castle, DE, USA). Lignin samples of about 5–10 mg were sealed in hermetic aluminum pans and tested under nitrogen atmosphere at a heating rate of 10 °C/min. Samples were first heated to 105 °C to eliminate interferences due to moisture and to erase the thermal history. Then the samples were cooled to 25 °C (0 °C for soluble fraction from diethyl ether) and reheated to 200 °C, for glass transition temperature (T_g_) determination.

## 3. Results and Discussion

### 3.1. Solubility Yields, Appearance, and Molecular Features of Lignin Fractions

Recovered hardwood kraft lignin from an industrial black liquor can be considered as a mixture of different lignin polymeric fractions. In this work, the one-step fractionation process was studied using nine organic solvents of different chemical nature to assess their ability to obtain more uniform lignin fractions with specific properties. The solvents were classified according to their chemical nature in addition to their dipole interactions (polar forces) and hydrogen bonding capacity based on the Hansen solubility parameter [21]. Methanol (δ_P_ = 12.3; δ_H_ = 22.3), ethanol (δ_P_ = 8.8; δ_H_ = 19.4), and 2-propanol (δ_P_ = 6.1; δ_H_ = 16.4) were used as polar protic solvents, while acetone (δ_P_ = 10.4; δ_H_ = 7.0), ethyl acetate (δ_P_ = 5.3; δ_H_ = 7.2), and dichloromethane (δ_P_ = 6.3; δ_H_ = 6.1) were selected as polar aprotic solvents. In addition, three different non-polar organic solvents such as diethyl ether (δ_P_ = 2.9; δ_H_ = 5.1), hexane (δ_P_ = 0; δ_H_ = 0), and petroleum ether (δ_P_ = 0; δ_H_ = 0) were used for kraft lignin fractionation. Aside from the chemistry of organic solvents, other aspects associated with their boiling points and environmental impact were also taken into consideration. Hence, from the selected solvents, three alcohols, acetone, and ethyl acetate were grouped as recommended solvents for chemical processes and suitable for industrial-scale implementation, while dichloromethane, diethyl ether, hexane, and petroleum ether were considered hazardous solvents which present strong restrictions for scale-up [22]. However, these last solvents were also included in the lab-scale study with the aim of gaining more insight into their influence on hardwood kraft lignin properties during fractionation. The yields of soluble and insoluble fractions from the one-step fractionation processes are presented in Figure 1. The solubility yields of HKL were linearly correlated to the polarity (δ_P_) of the solvents (R = 0.93) (Appendix A). High–medium solubility was observed in polar protic and aprotic solvents, which ranged from 40 to 86%, and from 30 to 77%, respectively. In case of alcohols, their hydrogen bonding capacity and polarity showed a significant influence on the fractionation yields. As can be observed, the solubility yield of kraft lignin in methanol, ethanol, and 2-propanol decreased as the number of carbons in the side chain increased, because longer aliphatic chains in alcohols reduce the polarity and the hydrogen bonding capacity of the alcohols [13,23]. Aprotic solvents such as acetone, ethyl acetate, and dichloromethane have lower hydrogen bonding capacity than alcohols; however, their polarity allowed the partial solubilization of HKL in the following order: acetone > ethyl acetate > dichloromethane. The ability of polar protic and aprotic solvents to solubilize high yield lignin fractions was previously reported by other authors [13]. With regard to non-polar solvents, crude HKL was insoluble in hexane and petroleum ether; however, diethyl ether was able to slightly solubilize around 11% of HKL, due to its cohesive energy related to dipole interactions and hydrogen bonding capacity. Wang et al. (2010) [24] fractionated Eucalyptus kraft lignin by sequential organic solvent extraction and evidenced an even lower solubilization yield (1.6%) of lignin in diethyl ether, demonstrating different behavior of lignin polymer even though they were from the same source. Other authors tested the solubility of lignin from bagasse isolated by acetosolv treatment in different organic solvents; the resulting isolated lignin was totally insoluble in diethyl ether. Therefore, although the trend of lignin to solubilize in organic solvents might be generalized, the solubility yield is highly dependent on the type of lignin and its origin [25].

Regarding the physical characteristics of lignin fractions, significant morphological differences as well as colour changes were observed between soluble and insoluble fractions (Appendix A). Soluble fractions resulted in a dark powder of irregularly shaped and sized particles with a caramel appearance. A similar appearance was found for insoluble fractions from ethanol and 2-propanol extractions. The rest of the insoluble fractions exhibited similar characteristics to the original lignin. The presence of fibres in insoluble fractions resulting from methanol and acetone fractionation processes was remarkable.

The molecular features of soluble and insoluble fractions were examined through gel permeation chromatography, analytical pyrolysis, and infrared spectroscopy, with the aim of evaluating the differences and similarities of fractionated lignins, and of studying the effectiveness of the organic solvents to obtain diverse types of lignins from the crude industrial kraft lignin. The average molecular weight and polydispersity indexes of both soluble and insoluble fractions are presented in Table 1.

The results demonstrated that a one-step fractionation method can be applied for refining HKL. Nevertheless, the molecular weight properties of HKL are greatly affected by the chemical nature of the solvent. The molecular weight and polydispersity of both soluble and insoluble fractions are clearly related to the polarity of the solvent. As expected, soluble fractions showed higher homogeneity and lower molecular weights than crude HKL, while remaining insoluble fractions, especially those which were obtained from polar protic and aprotic solvents, presented higher molecular weights and higher polydispersity values. Lignin is a highly polydisperse polymer, where lower molecular weight fractions have a higher solubility in organic solvents, while high molecular weight lignin has weak solubility in solvents. Moreover, higher carbohydrate chains linked to lignin can increase the hydrodynamic volume of lignin and, therefore, increased the apparent molecular weight of the lignin when it was measured using GPC [26]. Other authors have also evidenced high average molecular weight and heterogeneity for insoluble fractions coming from softwood kraft lignin fractionated by one-step processes with methanol, ethanol, propanol, acetone, methyl ethyl ketone, ethyl acetate tetrahydrofuran, and 2-butanone [11,13]. This effect was also observed by Tagami et al., (2019) [27] in insoluble fractions from hardwood and softwood kraft lignins fractionated by sequential solvent extraction. The present work shows that polar protic and aprotic solvents have a similar effect on the molecular weight properties of the studied lignin type. Solvents with lower polarity, but not totally apolar, such as hexane and petroleum ether, were able to extract more homogenous lignin fractions with lower molecular weights. These results indicate that the polarity of organic solvents is a key factor in controlling the molecular weight distribution of lignin during a refining process. In addition, it was observed that the remaining insoluble lignin from the one-step fractionation processes followed the same trend as for soluble fractions. Furthermore, analytical pyrolysis was used as a complementary tool for the characterization of lignin fractions, since volatile phenolic compounds released from the thermal degradation of lignin in the absence of oxygen provide useful chemical information about structural characteristics of the macromolecule [28,29]. In addition to phenolic compounds, during fast pyrolysis of lignin there were other pyrolytic products released such as furans and long-chain carboxylic acids and esters [17]. Furan derivatives derive from the degradation of polysaccharides and their fragmentation [30], and the presence of fatty acids is commonly associated with the origin and their resistance to thermo-chemical extraction processes [31]. However, since pyrolysis is not a quantitative method, the results from analytical pyrolysis evidenced the contribution of the released compounds, but were not suitable for the estimation of the composition of lignin samples [32].

The results of Figure 2 show that the area of total phenolic compounds for soluble fractions was in the range of crude HKL with values between 95–98%, while for insoluble fractions it was considerably lower, especially in the case of polar protic and aprotic solvents with higher polarity, which had high solubility yields of crude lignin. The high content of furan derivatives in these insoluble fractions was remarkable. When lignin was fractionated with organic solvents of high polarity, lignin fractions linked to carbohydrate moieties were not soluble, remaining in the insoluble fraction and contributing to their high molecular weight. Naturally, lignin and carbohydrates are chemically bonded in lignocellulosic materials forming a lignin–carbohydrate complex (LCC), which is stable during the alkaline kraft pulping process [33]. An increased carbohydrate content in high molecular weight lignins after fractionation processes was also observed by other authors while fractionating lignin both by organic solvents [13,34,35] and ultrafiltration methodologies [36]. These results indicate that one-step refining processes with high polar solvents such as acetone and methanol give a low purity insoluble lignin fraction. Table 2 presents the relative content (%) of phenolic-type compounds and their classification according to their origin and structural characteristics. Phenolic compounds released from fast pyrolysis were first classified into four categories according to their origin: phenol-type compounds (H), guaiacyl-type compounds (G), syringyl-type compounds (S), and catechol-type compounds (Ca). The syringyl/guaiacyl ratio (S/G) was calculated by dividing the sum of peak areas from syringyl units (including catechol derivatives) by the sum of peak areas of guaiacyl derivatives [37]. The results from the analytical pyrolysis and S/G ratio clearly reflected an increase in the content of guaiacyl-derived compounds and a reduction in syringyl-derived compounds in both soluble and insoluble fractions, except for the soluble fraction derived from diethyl ether. Other authors also observed a reduction in methoxylated aromatic compounds after lignin solubilization in organic solvents [14,38]. Recently, the interaction of organic solvents with methoxy groups of lignin was theoretically evidenced. Ponnuchamy et al. (2021) [39] studied the interaction of organic solvents with a lignin model compound and showed that several solvents trigger significant deviation in bond length of methoxyls. Additionally, this research proved that the hydrogen from methoxy carbon is a crucial site to form strong hydrogen bonding with organic solvents.

Released lignin-derived phenolic compounds were grouped based on the structural characteristics of their side chain as reported by Tagami et al., 2019 [27], and several differences were noticed between soluble and insoluble fractions. High content of compounds with non-substituted saturated chains such as 4-methylsyringol, 4-ethylsyringol, 4-methylguaiacol, 4-ethylguaiacol, 3-methoxycatechol, 4-propilsyringol and cresols, were identified in pyrograms of soluble fractions, especially in soluble fractions from isopropanol, ethyl acetate, and diethyl ether. The fast pyrolysis of insoluble fractions released less amount of these kind of phenolic compounds, indicating that soluble fractions contained more aliphatic saturated substructures than the initial HKL and remaining insoluble fractions. However, substructures with unsaturated side chains (carbon–carbon double bounds in α and β positions) such as vinylsyringol, vinylguaiacol, isoeugenol, and allylsyringol, were more abundant in insoluble fractions. This effect was particularly noted in insoluble fractions from high polar solvents such as methanol, ethanol, and acetone. Additionally, the higher content of allylsyringol compound found in insoluble fractions compared with crude HKL and soluble fractions was remarkable. Although double C=C bonds might be part of the initial lignin structure, this functional group can also be formed during the pyrolysis process [40]. Nevertheless, oxygenated substructures such as aldehydes and ketones derive from the original chemical structure when the lignin is pyrolyzed [29]. Regarding compounds with oxygenated functionalities in the side chains, in general, both soluble and insoluble fractions showed higher content of substructures with aldehyde and ketone functional groups, such as syringaldehyde, acetosyringone, and vanillin, than the initial HKL. It was especially evident for soluble fractions from methanol, ethanol, dichloromethane, and diethyl ether, and insoluble fractions from methanol, diethyl ether, and hexane. In addition, it was observed that solubilization of lignin in organic solvents of different nature generated soluble lignin fractions with a higher content of aromatic substructures with short aliphatic chains. In fact, the ratio between phenols with short and long side chains clearly showed significant structural differences between fractionated lignins. Additionally, infrared spectra were employed to identify variability in the chemical structure of studied lignin samples in relation to the type of solvent used for fractionation. The IR spectra were filtered (baseline, EMSC) and normalized (SNV), and then used for the PCA exploratory method, reducing the original spectra into 4–5 principal components (PC) including 99.82–99.90% of the cumulative variance (Table 3).

The PCA spectra of all fractions (soluble–insoluble–HKL) presented acceptable results (RMSEC: 0.424; RMSECV: 0.527) allowing the separation of the soluble and insoluble fractions when contrasting the scores of PC2–PC3–PC4 (Figure 3A). The scores plot of individual solvents is presented in the Supporting Information (Appendix A). The spectra of kraft lignin were within the range of the insoluble fractions and concretely in the quadrant shared by the nonpolar fractions plus dichloromethane, evidencing chemical similarities between them. However, the remaining insoluble fractions (methanol, ethanol, 2-propanol, and acetone) showed negative scores in PC2, indicating a chemical difference with the initial kraft lignin, insoluble fractions from the most non-polar solvents and soluble fractions. PC1 was no longer considered for discrimination because the loading plot had a similar pattern to that of the initial average spectra. Furthermore, the classification results in the PCA were focused on the 925–1650 cm^−1^ region, since in this region, changes in loadings were more clearly appreciated, and in general, the separation was more evident by reducing the overlapping of bands and outliers. In the analysis of the soluble fractions, the score plot of PC2–PC3 with less than 5% of spectral information displayed an interesting distribution and separation of samples according to the solvent nature, placing the kraft lignin as well as the sample fractionated with diethyl ether in an independent quadrant (Figure 3B). This evidenced chemical similarities between soluble fractions, but a clear differentiation with the initial kraft lignin and the fraction obtained from diethyl ether. In case of insoluble fractions, the distribution was more heterogeneous (Figure 3C), where samples resulting from the refining process with the most polar solvents (methanol, ethanol, 2-propanol and acetone) were furthest from the original kraft lignin, showing negative scores in PC2, while the rest of insoluble fractions were located close to HKL (positive PC2). The results from PCA supported the chemical analyses performed by GPC and analytical pyrolysis, and allowed for the classification of the lignin fractions according to their chemical characteristics in a simple and fast way.

### 3.2. Hygroscopic Properties of Lignin Fractions

Lignin has been always considered a hydrophobic compound when compared with the other polymers from lignocellulosic biomass such as hemicellulose and cellulose. However, lignin polymer contains a wide variety of functional groups able to interact with the vapor water molecules of the environment. Nevertheless, the water sorption–desorption behavior of lignin compound is still an unknown topic. This research work investigated the effect of the fractionation process on the hygroscopic properties of the resulting lignin fractions. Equilibrium moisture content of kraft lignin fractions during a sorption–desorption cycle calculated by saturated salt solution method is presented in Figure 4. The maximum moisture content in crude HKL at saturated conditions was around 18%. However, very different sorption–desorption patterns were observed for soluble and insoluble fractions. Soluble fractions reached up to 5–7% of moisture content at 95% relative humidity, while insoluble fractions showed higher water affinity, especially in the case where polar protic and aprotic solvents such as methanol and acetone were used for refining lignin. It was a clear trend regarding the insoluble fractions originating from organic protic and aprotic solvents; lower polarity of the solvent led to a less hygroscopic lignin fraction. It could be related to the higher contribution of carbohydrate derivatives of these fractions observed in the analytical pyrolysis, as the water absorption capacity of the insoluble fractions from polar solvent was proportional to the furan-related compounds found in the pyrograms. This indicates that higher sorption–desorption isotherms could be related to the high amounts of polysaccharides present in these fractions since hemicelluloses are the constituents in the cell wall of the plant with the highest affinity to water [41]. However, both the shape of the isotherm and the water absorption capacity of insoluble lignin fractions from apolar solvents were similar to crude HKL. As was reported in previous studies, lignin polymer exhibits sigmoid shape isotherms (type II) which are also common in wood and wood-derived compounds (macroporous materials) [41,42]. This type of isotherm is the result of the monolayer–multilayer adsorption mechanism of the water vapor molecules into the microstructure of the material [43] and are concave at low RH%, approximately linear in the intermediate region, and convex at high RH%. This pattern was observed in crude HKL and insoluble fractions, except in the insoluble fraction from 2-propanol, while soluble fractions obtained from the most polar protic and aprotic solvents showed almost linear shape isotherms with negligible hysteresis phenomenon. Hysteresis is a result of the difference between equilibrium moisture content in sorption and desorption at the same relative humidity. As can be observed, insoluble fractions evidenced higher hysteresis than soluble fractions indicating their lower capacity to release the adsorbed water during a desorption process. According to previously reported studies, the hygroscopic property of materials is related to the availability of sorption sites to bond with the water vapor molecules. In the case of lignin, hydroxyl and carboxylic groups were recently recognized as active water sorption sites [44,45]. Several authors reported that lignins with lower molecular weight contained higher hydroxyl and carboxylic groups in their chemical structure [12,14,46]. Additionally, our results proved that the soluble fractions showed higher total phenolic content than insoluble fractions (Table 4); however, they exhibited less affinity to water vapor. Therefore, based on our results and according to literature, the authors support the theory that the hygroscopicity and hysteresis phenomenon of lignin polymer is influenced by the chemical structure, but it is especially dominated by the microstructure of the polymer, which plays an essential role in hampering or favoring the availability of these functional groups and their interaction with water vapor molecules.

### 3.3. Total Phenolic Content and Antioxidant Capacity of Lignin Fractions

The results of the Folin–Ciocalteau assay and the antioxidant capacity against the DPPH radical are included in Table 4. Total phenolic content of lignin was expressed as gallic acid equivalents, and the antioxidant capacity was expressed as the concentration required for 50% inhibition of the radical (IC_50_). The TPC is highly correlated to the antioxidant property of lignins since the DPPH radical can be reduced with the addition of a hydrogen atom from phenolic hydroxyl groups of lignin (hydrogen atom transfer reaction mechanism) and subsequent formation of the phenoxy radical. Nevertheless, the radical power of lignin not only depends on the formation of the phenoxy radical but also on its stability. Chemical functionalities such as methoxy groups, conjugated double bonds, the existence of CH_2_ in the α-position of the side chain as well as short side chains attached to the aromatic ring have a positive effect on the antioxidant capacity of lignin polymer [27,37,40]. In this research, the results evidenced that soluble fractions showed higher TPC than original lignin and insoluble fractions. Moreover, it was observed that the TPC increased inversely with the molecular weight for both soluble and insoluble fractions, which is in agreement with previous studies [16]. Based on the DPPH assay, lignin fractions with improved antioxidant capacity by the one-step refining process using organic solvents can be obtained. Soluble fractions showed higher values of (ArC_1_ + ArC_2_)/ArC_3_ than insoluble lignins, which were previously correlated with higher antioxidant activity of the polymer [37]. Moreover, the radical scavenging activity of soluble fractions was improved in the following order based on the nature of the used solvent: non-polar > polar aprotic > polar protic. However, insoluble fractions exhibited poor antioxidant capacity, especially those resulting from fractionation with the most polar solvents, due to their high molecular weight and impurities.

### 3.4. Thermal Properties of Lignin Fractions

The thermal properties of lignin polymer are very important for applications in the polymeric field and carbon-based materials production [10,16] as well as in obtaining high-value added chemicals through thermochemical processes. Moreover, the decomposition profile of lignin fractions can provide information on the lignin chemistry based on the thermal breakdown of its structure [47]. In this work, high-resolution TGA technique was used, which differs from conventional thermal analyses in that the heating rate of the sample is dynamically and continuously modified in response to changes in the rate of decomposition of the sample, improving the weight change resolution and optimizing the time of analysis. The thermogravimetric (TG) and first derivative (DTG) curves of lignin samples under nitrogen atmosphere are presented in Figure 5. The initial degradation temperature (T_10%_), the maximum weight loss temperatures (T_max_), and the temperature where 50% of lignin was decomposed (T_50%_) together with the char residue at 800 °C and ash content are summarized in Appendix A. The results clearly evidenced that the nature and polarity of organic solvents have a great influence on the thermal properties and degradation profile of lignin fractions from a one-step refining process. Moreover, significant differences were observed between soluble and insoluble fractions in terms of thermal stability. The thermal stability is affected not only by the molecular weight of lignin but also by the chemical structure (linkages, functional groups, and degree of condensation) [12]. The thermal degradation of lignin polymer generally occurred from 150 to 800 °C through various stages of degradation which were attributed to the cleavage of different bonds in the lignin molecule. Aside from the first small weight loss below 100 °C related to dehydration, most of the research works reported two main degradation peaks on the first derivative; however, the high-resolution thermal analysis carried out in this study resolved other degradation stages of studied lignin fractions. The main decomposition regions occurred in the temperature ranges of 120–180 °C, 180–250 °C, 280–380 °C, and above 400 °C. After dehydration, the first degradation step, which occurred between 180 and 250 °C, was attributed to the cleavage of bonds with low activation energy such as β- and α- aryl ether linkages [48]. The weight loss in this range was clearly observed in soluble fractions, while in crude HKL and some insoluble lignin fractions it appeared as a shoulder. At higher temperatures, lignin degradation is due to stable bond fragmentation such as carbon–carbon linkages between aromatic units, cracking of aliphatic side-chains, and cleavage of functional groups [10,49]. The main weight loss stage in all lignins occurred in the temperature range between 280 and 380 °C and was centered around 320–350 °C and 280–330 °C, for soluble and insoluble fractions, respectively. Unexpectedly, soluble fractions with lower molecular weight presented higher thermal stability than their insoluble counterparts, which showed relatively lower thermal stability, especially those from polar solvent fractionation processes. Generally, lignin fractions with higher Mw have more β-O-4 linkage structures and lower functional groups content [50]. This, together with their lower quality (reflected by Py-GC-MS), could explain the low thermal stability of these fractions, especially in the case of insoluble fraction from methanol and acetone. In the case of soluble fractions isolated with polar protic solvents, the less polar solvent used, the higher the observed T_max_. However, this trend was not evidenced for polar aprotic solvents. After the main degradation step, a shoulder was manifested around 380–480 °C in both soluble and insoluble fractions which corresponded to the cleavage of β-β bonds, and to condensation and polymerization reactions, respectively [48,51]. In addition, the existence of a clear degradation peak at high temperatures (700–800 °C) in insoluble fractions from polar solvents was evidenced. Araújo et al. 2020 also detected a thermal degradation peak at 702 °C in an insoluble fraction resulting from a sequential fractionation of hardwood kraft lignin with five organic solvents, and confirmed that this degradation stage was associated with inorganic compounds from the kraft pulping process (Na and S) which were not soluble in organic solvents and were retained in the insoluble fractions [52]. The char residue at 800 °C increased with increasing molecular weight for each group of solvents for soluble lignin fractions. Therefore, solvents with lower polarity lead to a lignin fraction with lower non-volatile residue after thermal processing, which could have a positive effect for fuels and low molecular weight aromatic chemical compounds production. This is related to the content of functional groups such as hydroxyls and methoxyls, which usually are higher in low molecular weight lignin fractions [52]. Other authors observed a similar trend [10,12,51]. However, the opposite trend was noticed for insoluble lignin fractions. In addition, the ash content was determined under oxygen atmosphere at 800 °C. Soluble fractions showed less ash content than crude HKL, and it was lower as the polarity of the solvent decreased. However, insoluble fractions evidenced very high ash content since most of the inorganic material was insoluble in organic solvents, becoming part of the insoluble fraction composition. Other authors also confirmed that the remaining insoluble fractions usually had the highest level of inorganics [35]. Therefore, it was demonstrated that the nature of the solvent has an important effect on the purity of the resulting lignin fractions.

Differential thermal analysis was performed to determine the glass transition temperature (T_g_) of lignin fractions (Table 5). The DSC curves are displayed in Appendix A of the Supporting Information. The results demonstrated that the glass transition temperature was highly affected by the molecular weight properties of soluble fractions and, hence, by the nature of the organic solvents used for refining. With polar protic and aprotic solvents, the T_g_ decreased with the molecular weight, and the fractionation using diethyl ether resulted in a soluble lignin fraction with very low glass transition temperature (52.9 °C). Previous studies reported that insoluble fractions in general have higher T_g_ than their soluble counterparts [11,53]; however, the authors could not find the same trend in this study, especially for those insoluble fractions obtained from the most polar solvents.

## 4. Valorization Perspectives of Lignin Fractions

The oldest lignin valorization approach, which is considered to be a low-value application, is heating fuel and energy obtention through combustion. However, research and industrial interest are centered on the production of high-value added chemicals such as aromatic monomers and oligomers through thermochemical and biochemical fragmentation technologies, and application of lignin as a macromolecule in different fields such as polymeric, cosmetic, and biomedical [54]. Generally, the performance of kraft lignin in various target applications when used without any conversion treatment is usually poor. To date, several strategies have been developed to overcome the technical limitations of lignin and achieve the desired functioning of this biopolymer in a final material or product. Refining of lignin using organic solvents produces controlled fractions with relatively consistent quality in terms of molecular weight and properties, allowing engineers to predict lignin behavior in a specific application. Based on the results of this study, this section aims to propose potential applications for fractionated lignin samples in the polymeric form. General comparison of the properties of the fractionated lignin samples with the original hardwood kraft lignin isolated from industrial black liquor is presented in Table 6. Application of lignin as a macromolecule has been extensively studied in recent decades in the polymeric field. Lignin has been considered as a green alternative to reduce the fossil component in thermoplastics, providing new or improved properties to the final material, but also as a renewable substitute of phenol and polyols in the synthesis of phenol-formaldehyde resins and polyurethanes, respectively. High molecular weight and polydisperse lignin usually have low reactivity towards formaldehyde or isocyanate; however, one-step solvent fractionation could be used as an efficient strategy to increase the reactivity and the availability of hydroxyl groups [55]. For instance, the soluble fraction resulting from extraction with diethyl ether which presented very low Mw and polydispersity, may be suitable for these kinds of applications. Other soluble fractions, such as those obtained from 2-propanol, ethyl acetate, and dichloromethane also exhibited lower Mw, polydispersity, and higher TPC than original HKL, showing potential for the synthesis of PF and PU. In addition, due to its chemical features, lignin can act as an antioxidant, UV, and thermal stabilizer in the polymer matrix. High content of phenolic hydroxyls has a positive effect on the antioxidant and UV radiation absorbing capacity of lignin; however, a good compatibility between the polymer matrix and lignin, and uniform dispersion are essential requirements for a suitable response of lignin against oxidation induced by UV and thermal processing [56,57]. Functionalization of hydroxyl groups generally improves compatibility with apolar polymers such as polypropylene, polyethylene or polylactic acid, but antioxidant properties and UV stabilization capacity are usually reduced. Therefore, soluble fractions obtained with lower polarity solvents such as 2-propanol, dichloromethane, and diethyl ether could be suitable as an additive in thermoplastics because of their relatively low molecular weight and narrow polydispersity, in addition to low ash content, low T_g_, acceptable thermal stability, and improved antioxidant capacity. In the case of lignin as a carbon precursor for the production of carbon fibers, having high char yield and low impurities such as salt, water and carbohydrates are favorable [12,58]. Moreover, high molecular weight often results in a higher mechanical performance of fibers. It could be expected that insoluble fractions from polar solvents which presented high Mw would be the most suitable fractions for such kind of applications; however, those fractions contained very high ash content and contamination of carbohydrates. Therefore, for use as a source of carbon fiber or activated carbon, fractions from diethyl ether and hexane would be most recommended.

## 5. Conclusions

In this study, nine organic solvents of different chemical nature were studied for refining hardwood kraft lignin by a one-step fractionation process. Fractionation conditions resulted in soluble and insoluble fractions with different chemical and functional properties. The nature and polarity of the solvents were the key factors in controlling the molecular weight distribution of fractionated lignins and showed a clear relationship with the resulting properties and quality of both soluble and insoluble fractions. Regarding the solvents, the authors found that acetone showed a similar effect on hardwood kraft lignin as to the studied polar protic solvents, especially to methanol. Chemometric analysis of infrared spectra of fractionated lignins supported the chemical analysis and allowed for a quick classification of soluble and insoluble fractions according to their similarities and differences. Soluble fractions presented higher homogeneity, reduced hygroscopicity and glass transition temperature, improved antioxidant capacity, lower char residue at 800 °C, and less ash contamination than the original hardwood kraft lignin. However, the lower quality of remaining insoluble fractions was proved, especially those from polar solvents such as methanol, ethanol, 2-propanol, and acetone, which presented lower purity, and reduced ability to act as an antioxidant compound and thermal stabilizer.

## Figures and Tables

**Figure 1 polymers-14-02363-f001:**
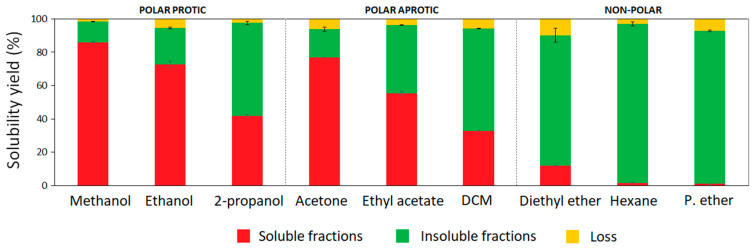
Yields of soluble and insoluble lignin fractions from one-step extraction processes.

**Figure 2 polymers-14-02363-f002:**
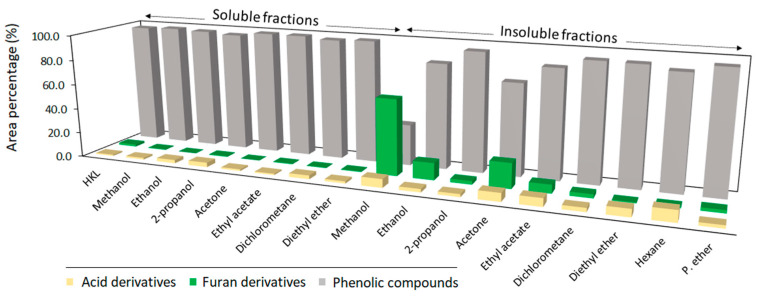
Relative content of identified pyrolysis products according to their origin.

**Figure 3 polymers-14-02363-f003:**
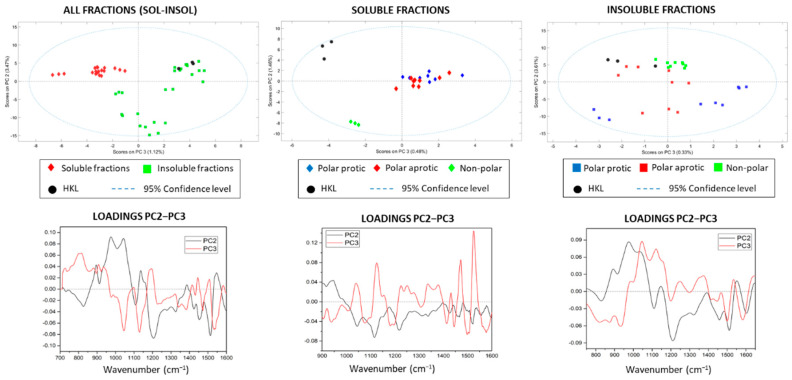
FTIR analysis, scores, and loadings of all samples, and in the soluble and insoluble fractions.

**Figure 4 polymers-14-02363-f004:**
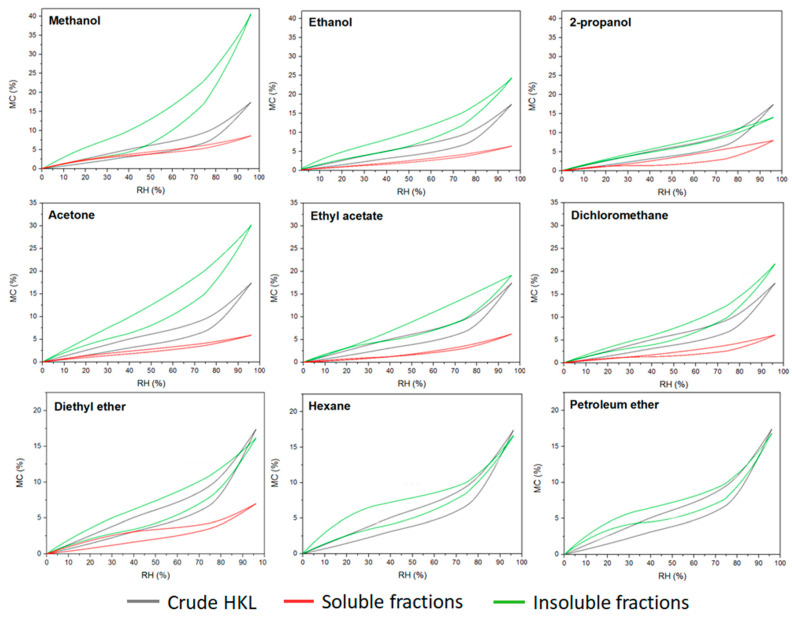
Experimental sorption–desorption isotherms of lignin fractions determined by saturated salt solution method.

**Figure 5 polymers-14-02363-f005:**
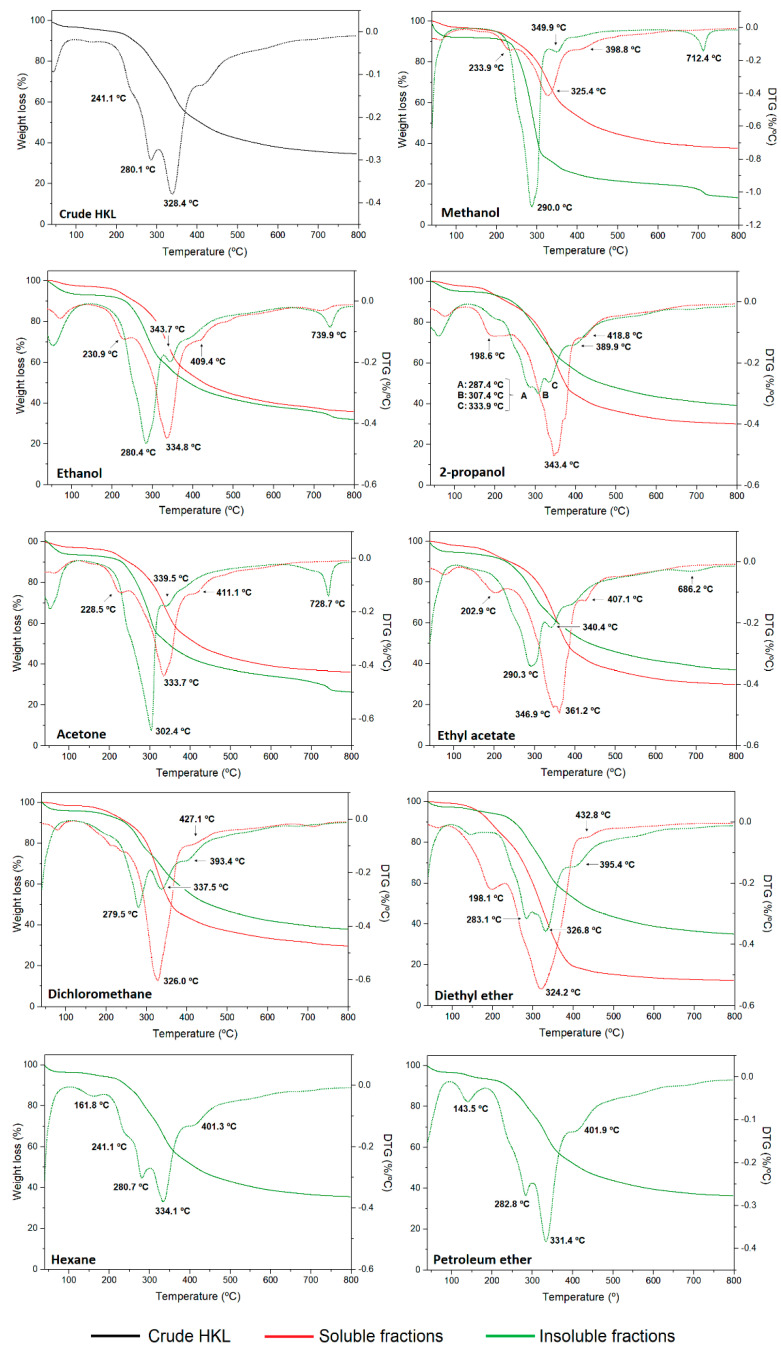
TG and DGT curves of crude HKL, soluble and insoluble fractions.

**Table 1 polymers-14-02363-t001:** Molecular weight-average (Mw), number-average (Mn) and polydispersity (PDI) of HKL and isolated lignin fractions.

	Soluble Fractions			Insoluble Fractions		
	Mn (g/mol)	Mw (g/mol)	PDI (Mw/Mn)	Mn (g/mol)	Mw (g/mol)	PDI (Mw/Mn)
Crude HKL	669	2477	3.7	-	-	-
Methanol	651	2272	3.5	3110	1,4370	4.6
Ethanol	633	1982	3.1	1762	8492	4.8
2-propanol	587	1245	2.1	1144	4302	3.8
Acetone	661	2251	3.4	1670	1,0384	6.2
Ethyl acetate	561	1430	2.6	1272	5568	4.4
Dichloromethane	562	1195	2.1	1158	4453	3.8
Diethyl ether	412	563	1.4	759	2894	3.8
Hexane	-	-	-	676	2569	3.8
Petroleum ether	-	-	-	675	2614	3.9

**Table 2 polymers-14-02363-t002:** Relative content (%) of phenolic-type compounds according to their structural characteristics and S/G ratio.

Sample		Phenolic Compounds: Structure of the Side Chain						S/G
		Non-Substituted Saturated Chains (%) ^a^	Unsaturated Side Chains (%) ^b^	Oxygenated Groups in the Side Chains (%) ^c^	Short Side Chain (%) ^d^	Long Side Chain (%) ^e^	(ArC_1_ + ArC_2_)/ArC_3_ ^f^	
HKL		29.9	6.7	1.4	36.1	2.1	17.1	4.8
Methanol	S	31.2	4.5	3.0	37.0	1.6	22.9	2.8
	I	20.4	9.3	35.1	46.4	3.9	11.9	3.5
Ethanol	S	30.8	5.7	6.5	40.0	3.3	12.0	3.4
	I	25.6	10.9	2.3	35.5	5.5	6.5	2.7
Isopropanol	S	36.7	6.4	1.9	41.6	3.0	14.1	3.3
	I	27.9	7.5	3.5	35.4	3.6	9.9	3.1
Acetone	S	34.5	6.6	2.0	40.7	3.4	11.8	3.0
	I	16.5	9.0	5.1	27.2	3.7	7.4	3.1
Ethyl Acetate	S	45.8	4.8	1.9	49.6	2.9	17.3	2.6
	I	22.8	6.7	4.0	28.2	5.3	5.3	2.8
Dichloromethane	S	37.1	5.4	7.2	46.0	3.9	11.9	3.1
	I	23.0	7.4	3.7	30.3	3.5	8.8	3.2
Diethyl ether	S	40.1	3.5	3.6	45.7	2.3	20.2	5.2
	I	21.7	7.4	18.6	42.8	4.7	9.0	4.3
Hexane	S	-	-	-	-	-	-	-
	I	25.0	6.6	9.6	37.2	4.9	7.5	3.7
Petroleum Ether	S	-	-	-	-	-	-	-
	I	27.8	8.6	3.7	35.5	4.1	8.7	3.0

^a^ Short and propanoid side chains attached to the aromatic ring with no unsaturated bonds; ^b^ Compounds with unsaturation in C_α_ = C_β_ and C_β_ = C_γ_ positions; ^c^ Phenolic compounds containing aldehyde and ketone groups in the side chain; ^d^ Short side chains refer to 1 and 2 carbon side chains; ^e^ Propanoid side chains; ^f^ Ratio between phenols with short and long side chains; S: soluble and I: insoluble.

**Table 3 polymers-14-02363-t003:** Summary of the PCA results in all fractions and in separated groups.

Included Data	Data Preprocessing	PCs	% Cumulative Variance	RMSEC	RMSECV
All fractions	Baseline–EMSC–SVN	5	99.82	0.0424	0.0527
** *Analysis of individual class groups* **					
Insoluble fractions	Baseline–EMSC–SVN	4	99.90	0.0310	0.0414
Soluble fractions		4	99.79	0.0457	0.0703
*PCs = principal components*					

**Table 4 polymers-14-02363-t004:** Total phenolic content (TPC) and efficient concentration (IC_50_) of fractionated lignin samples.

	Soluble Fractions		Insoluble Fractions	
	TPC (µg GA/mg Lignin)	IC_50_ (µg/mL)	TPC (µg GA/mg Lignin)	IC_50_ (µg/mL)
Crude HKL	355.5 ± 3.4	11.4 ± 0.4		
Methanol	377.3 ± 1.4	9.9 ± 0.4	61.9 ± 3.0	46.4 ± 4.2
Ethanol	389.4 ± 2.1	9.9 ± 0.6	182.8 ± 1.9	18.7 ± 0.1
2-propanol	411.9 ± 8.6	9.5 ± 1.0	299.5 ± 0.8	13.1 ± 0.8
Acetone	388.1 ± 0.1	9.7 ± 0.0	163.2 ± 2.6	19.8 ± 0.0
Ethyl acetate	388.6 ± 3.4	8.4 ± 0.0	291.7 ± 1.4	9.7 ± 0.4
Dichloromethane	449.0 ± 7.4	8.1 ± 0.3	325.5 ± 3.8	11.9 ± 0.9
Diethyl ether	407.6 ± 5.4	8.5 ± 0.8	365.1 ± 4.4	9.5 ± 0.5
Hexane	-	-	361.3 ± 0.2	9.9 ± 0.5
Petroleum ether	-	-	345.2 ± 1.3	10.1 ± 0.5

Note: The IC_50_ of ascorbic acid was 4.60 ± 0.1 µg/mL.

**Table 5 polymers-14-02363-t005:** Glass transition temperature of lignin fractions from hardwood kraft lignin.

	Soluble Fractions	Insoluble Fractions
	T_g_ (°C)	
Crude HKL	133.2	
Methanol	122.6	96.9
Ethanol	113.8	99.4
2-propanol	84.8	86.6
Acetone	114.9	93.5
Ethyl acetate	89.2	138.3
Dichloromethane	79.2	173.1
Diethyl ether	52.9	151.8
Hexane	-	137.0
Petroleum ether	-	127.0

**Table 6 polymers-14-02363-t006:** Summary table of all tests performed on lignin fractions.

	Homogeneity ^a^	Hygroscopic Stability	Antioxidant Capacity	Thermal Stability ^b^	Ash Content
* **Soluble fractions** *					
Methanol	+	+	+	=	+
Ethanol	+	+	+	+	++
2-propanol	++	+	+	+	++
Acetone	+	+	+	=	++
Ethyl acetate	+	+	++	+	++
Dicloromethane	+	+	++	=	++
Diethyl ether	++	+	++	=	++
* **Insoluble fractions** *					
Methanol	−	−	−	−	−
Ethanol	−	−	−	−	−
2-propanol	−	=	−	−	−
Acetone	−	−	−	−	−
Ethyl acetate	−	=	+	−	−
Dicloromethane	−	=	=	−	−
Diethyl ether	=	=	+	=	=
Hexane	=	=	+	=	=
Petroleum ether	=	=	+	=	=

^a^ Reduced Mw distribution and polydispersity; ^b^ According to T_max_ value; − Deteriorate; = No improvements; + Improvements; ++ Remarkable improvements.

## Data Availability

Not applicable.

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
