# Peer review of "One-Step Lignin Refining Process: The Influence of the Solvent Nature on the Properties and Quality of Fractions"

_polymers, 2022, doi:10.3390/polym14122363_

Round 1

Reviewer 1 Report

The authors investigated the “One-step Lignin Refining Process: the Influence of the Solvent 2 Nature on the Properties and Quality of Fractions”.  In this work, the authors have done a thorough dissolution study of lignin soluble and insoluble fractions using nine organic solvents. These fractions were also characterized using several notable techniques. However, looking at the data and discussion, several points are missing, and some details comments are below.

1.      What was the molar mass range of monodispersed polystyrene used to determine the molar mass of lignins?

2.      What was the weight of lignin conditioned to determine the hygroscopic properties, and also add the capacity of the sealed container along with the volume of saturated solution used?

3.      For the thermal analysis 5-10 g of lignin was used, make sure there is no typo?

4.      Authors found similar molar masses of soluble and insoluble fractions in this work to reported elsewhere. That's excellent, but what are the reasons for having similar molar masses of soluble fractions to HKL and high molar masses of insoluble fractions. Authors should explain rather than just add the references.

5.      Figure 2 shows the excellent results for the selective separation of polysaccharides from HKL. It would be an excellent discussion if the authors can provide the composition of HKL, and correlate that with the methanol insoluble fractions.

6.      Wondering to see the results of the high S/G ratio of methanol soluble relative to insoluble. The authors stated that methoxy groups formed strong hydrogen bonding and should be yielded in more soluble fractions in methanol. Whereas the pyrolysis trend is reversed in the case of methanol. This statement looks controversial to the author’s pyrolysis results.

7.      Figure 4 needs more clarity, as HKL contains polysaccharides, and different amounts of polysaccharides are in soluble and insoluble fractions. Methanol and acetone insoluble fractions yielded higher sorption-desorption isotherms, because they have high amounts of polysaccharides (Fig. 2), and polysaccharides have more affinity with water. So, the sorption-desorption isotherms of lignin methanol and acetone insoluble fractions need to be deleted or need more discussion.

8.      Why TPC of methanol insoluble fraction is substantially low relative to other insoluble fractions?

9.      What could be the reasons for the more weight loss of methanol insoluble fractions ~ 300 oC relative to other fractions?

Author Response

Thank you very much for your time in reviewing our manuscript as well as for your valuable comments and critics. We have deeply discussed your suggestions and modified the manuscript accordingly. Our detailed, point-by-point responses are given in the pdf (in blue colour), whereas the corresponding revisions were highlighted in red colour throughout the text. Additionally, we have carefully revised the manuscript to minimize typographical, grammatical, and bibliographical errors. We do sincerely hope that our modifications (answers) fulfil your requirements.

Reviewer 2 Report

The manuscript can be published in the light of the observations included in the attached PDF.

Author Response

Thank you very much for your time in reviewing our manuscript as well as for your valuable comments and critics. We have deeply discussed your suggestions and modified the manuscript accordingly. Our detailed, point-by-point responses are given in a pdf (in blue colour), whereas the corresponding revisions were highlighted in red colour throughout the text. Additionally, we have carefully revised the manuscript to minimize typographical, grammatical, and bibliographical errors. We do sincerely hope that our modifications (answers) fulfil your requirements.

Round 2

Reviewer 1 Report

The authors responded to the questions with satisfactory answers and included the discussion in the revised manuscript. Therefore, I would suggest considering this manuscript for publication in Polymers.